# The Dietary Approach to the Treatment of the Rare Genetic Tubulopathies Gitelman’s and Bartter’s Syndromes

**DOI:** 10.3390/nu13092960

**Published:** 2021-08-26

**Authors:** Francesco Francini, Laura Gobbi, Verdiana Ravarotto, Silvia Toniazzo, Federico Nalesso, Paolo Spinella, Lorenzo A Calò

**Affiliations:** 1Department of Medicine (DIMED), Clinical Nutrition, University of Padova, 35128 Padova, Italy; francesco.francini@aopd.veneto.it (F.F.); silvia.toniazzo@libero.it (S.T.); paolo.spinella@unipd.it (P.S.); 2Nephrology, Dialysis and Transplantation Units, Department of Medicine (DIMED), University of Padova, 35128 Padova, Italy; laura.gobbi_01@aopd.veneto.it (L.G.); verdiana.ravarotto@gmail.com (V.R.); federico.nalesso@unipd.it (F.N.)

**Keywords:** Gitelman’s syndrome, Bartter’s syndrome, nutritional therapy, sucrosomial magnesium, potassium supplements

## Abstract

Gitelman’s (GS) and Bartter’s (BS) syndromes are rare, inherited autosomal recessive tubulopathies characterized by hypokalemia, metabolic alkalosis, renal sodium, chloride, and potassium and magnesium-wasting. While the treatment based on potassium, sodium, chloride, and magnesium supplementation in addition to other pharmacologic options are widely established, recommendations about the dietary approach to GS and BS still remain generic. In this review we focus on the dietary strategies to increase sodium, potassium, and magnesium intake in GS and BS patients. Potassium and magnesium-rich foods and supplements are considered together with those that may reduce through different mechanisms the potassium and magnesium plasma level. Magnesium supplementation is often poorly tolerated, causing abdominal pain and diarrhea in most patients. New formulations using liposome and, in particular, sucrosomial technology have been recently proposed for magnesium supplementation in order to increase magnesium supplement tolerability and intestinal absorption. The dietary approach to GS and BS may be very important in the therapeutic approach to these syndromes. Due to the relevance of the dietary approach to these syndromes, a nutritional counseling should always be recommended and the nutritionist should join nephrologists in the follow-up of GS and BS patient care.

## 1. Introduction

### Gitelman’s Syndrome and Bartter’s Syndrome

Gitelman’s syndrome (GS) [1] and Bartter’s syndrome (BS) [2] are two salt-losing tubulopathies characterized by hypokalemic metabolic alkalosis with high activation of the renin–angiotensin–aldosterone system, with high renin and aldosterone levels yet hypotension or normotension. Both the diseases are recessively inherited, caused by inactivating mutations in genes’ encoding channels or cotransporters expressed in the thick ascending limb or in the distal convoluted tubule which engenders several subtypes of the diseases with different manifestations and severity.

The presence of both hypocalciuria and hypomagnesemia, in addition to hypokalemia, are highly predictive for the clinical diagnosis of GS, whereas five different forms have been identified, characterized by hypokalemic and hypochloremic metabolic alkalosis, intravascular volume depletion because of renal salt wasting, hyperreninemia and hyperaldosteronism, and low to normal blood pressure depicting BS types 1–5 [3,4]. A further GS-like phenotype, including hypomagnesemia and hypocalciuria, has been also associated with mutations in the CLCNKB gene encoding the chloride channel ClC-Kb, the cause of Bartter’s syndrome type III [3] (Figure 1).

GS and BS syndromes usually differ in terms of the specific onset timing: while GS is usually detected during adolescence or adulthood and may be asymptomatic or associated with mild or non-specific symptoms such as muscular weakness, fatigue, salt craving, thirst, nocturia, or cramps, BS is revealed in childhood or even as an antenatal/neonatal form. However, severe manifestations of GS have been described, such as early onset before 6 years of age, leading to growth retardation, chondrocalcinosis, tetany, rhabdomyolysis, seizures, and ventricular arrhythmia [3].

A high phenotypic variability has been documented in genetically confirmed GS/BS patients and may be used to find an explanation for the involvement of a combination of genotype, sex, modifier genes, compensatory mechanisms, as well as environmental factors or dietary habits.

The therapeutic approach for most patients with GS/BS is based on supplements of sodium chloride, potassium chloride, oral magnesium (for GS patients), and fluids with individual adjustments based on symptoms, tolerability, severity of the disease, age of the patient, and the glomerular filtration rate. In addition, non-steroidal anti-inflammatory drugs are largely used in most BS patients in particular during the first years of life. The use of other therapies, such as potassium-sparing diuretics, angiotensin converting enzyme inhibitors, and angiotensin receptor blockers, are also included in the pharmacologic treatment [3,4].

Although food is the main source of minerals, dietary interventions other than the encouragement to consume potassium-rich and salty foods are not considered in detail in GS and BS treatment and the recommendation to consume potassium and magnesium supplements is almost always generic [3,4].

While the treatment based on potassium, sodium, chloride, and magnesium supplementation, in addition to other pharmacologic options, are widely established, recommendations about the dietary approach to GS and BS still remain generic. In this review, we focus on the dietary strategies to increase sodium, potassium, and magnesium intake in GS and BS patients with potential benefits for the electrolyte alterations of these patients.

## 2. Dietary Approach to Gitelman’s Syndrome

### 2.1. Sodium and Chloride

The dietary approach recommended to counteract the sodium and chloride losses is an ad libitum salt diet supplemented with sodium chloride tablets. Slow-release sodium tablets could be used at the dose of 2.4–4.8 g per day in four divided doses [5]. The salt craving showed by many patients can help to introduce salty foods or salt supplementation.

There are currently no clinical trials testing the benefits of sodium replacement therapy in GS patients [2]. It appears intuitive that an elevated salt intake increases renal tubular sodium loading and therefore in the RAAS suppression, the reduction of alkalosis, as well as helping to raise the potassium and chloride plasma level.

The clinical picture of BS, although much more severe, presents clinical characteristics virtually indistinguishable from GS [4] and its dietary treatment largely shares the same interventions. However, early onset, hypochlorhydria, and more severe symptoms require further measures. Post-natal treatment supplementation with sodium chloride represents the main treatment to restore extracellular volume and improve electrolyte abnormalities. At least 5–10 mmol/kg/d (291–584 mg/d) has been recommended [4,6].

Even if the literature lacks indications regarding this topic, in young patients, requiring high doses of sodium and potassium chloride enteral nutrition through tube feeding or percutaneous gastrostomy growth could be considered in order to maximize caloric intake and assure optimal growth.

In late childhood patients, the increase of dietary salt intake due to salt craving is important. In this case, salt addition to foods is preferable considering the salty foods available include processed and smoked meat, chips, snacks, and salty bakery goods whose prolonged use is unhealthy and could favor degenerative diseases [7].

Salt supplementation must be avoided in BS type 1 and type 2 patients with secondary forms of nephrogenic diabetes insipidus. The solute load, due to the high salt intake, in fact, could worsen polyuria and induce hypernatriemic dehydration [4,8].

### 2.2. Potassium

Hypokalemia is one the more frequent and crucial manifestations of GS. A recent consensus on GS suggested that a reasonable target for potassium may be 3.0 mmol/L [2]. Potassium-rich foods such as fruits, vegetables, meat, poultry, and fish are the main natural sources of this element (Table 1) and their regular intake can be recommended to increase dietary potassium.

There are some differences between meats, fruits, and vegetables as sources of potassium. Meat increases the acid load due to the high content of organic sulfur in its proteins [9], while fruit and vegetable intake is associated to a net base production due to their content of organic acid anions [10]. Moreover, the carbohydrate content of fruits and vegetables stimulates insulin secretion, favoring intracellular entry of potassium. It should also be considered that potassium losses from cooking of high potassium-rich foods can be significant [11]. The main preparation method that loses potassium from many foods is through boiling in water. However, if the cooking water is consumed, all the potassium is introduced. Steaming vegetables could be used instead of boiling as steamed foods retain potassium, which is not lost in the cooking water. Some potassium-rich foods, including chocolate, peanuts, pumpkin seeds, fruit juices, and cereals, containing large amounts of carbohydrates or fat should be limited in order to prevent an excessive intake of calories.

When potassium replacement is administered as oral supplement, it should be given in the form of chloride, which is the main anion lost in the urine of GS patients, contributing to alkalosis. However, other formulations are also available. Potassium supplementation may cause serious side effects including vomiting, diarrhea, and gastric ulcers. To minimize these side effects, potassium supplements must be taken on a full stomach and slow-release formulation should be preferred. A recent consensus suggests a starting dose of ≥40 mmol KCl (1–2 mmol/kg in children) in divided doses throughout the day, with 3.0 mmol/L as the target for the potassium plasma level [2]. 

When patients do not tolerate oral potassium or when the target cannot be reached, intravenous KCl should be administered [2]. The concentration of potassium for intravenous administration via a peripheral vein should not exceed 40 mmol/L at a maximum rate of 10 mmol/h, as higher concentrations can cause phlebitis and pain. In severe cases of hypokalemia, KCl solution up to 80 mmol/L at a maximum rate of 20 mmol/h should be given via a central venous route. In this case, an infusion pump may be required as well as plasma potassium and cardiac monitoring which is recommended [2].

Recommendations about dietary potassium reported for GS patients should be applied also for BS patients [4]. Although the optimal plasma potassium level is not exactly known, 3.0 mmol/L is recommended as a reasonable target level [4]. As BS patients lose large amounts of chloride, as suggested for GS patients, potassium chloride is the salt that should be used in order to counteract hypochloremia and metabolic alkalosis.

With the potassium assumption, urinary losses shortly increase, causing a fast reduction of the plasma potassium level. Therefore, supplements of potassium chloride should be taken several times a day and not on an empty stomach due to the risk of gastric damage. Slow-release formulations are preferred in order to reduce adverse effects such as vomiting and diarrhea, which can worsen water and electrolyte losses.

Similarly to GS patients, a potassium rich diet is recommended also for BS patients. However, many potassium-rich foods contain large amounts of carbohydrates (fruit juices, potatoes, sweet potatoes, bananas, dried apricots, etc.). These foods should be eaten with moderation as they can induce an important insulin secretion with the consequent migration of potassium into the cells and a further reduction of the plasma potassium level [12].

### 2.3. Magnesium

The exact mechanisms of hypomagnesemia as well as hypocalciuria in GS, is still uncertain. Hypomagnesemia in GS is assumed to be a consequence of the inactivation of NCC in the distal convoluted tubule (DCT) with a mechanism similar to thiazides. In NCC-knockout mice, an animal model of GS, Nijenhuis et al. demonstrated a downregulation of the epithelial Mg^2+^ channel transient receptor potential channel subfamily M, member 6 (TRPM6), which explains the decreased active transport of Mg^2+^ in the DCT [13].

However, data suggest that magnesium-wasting is the primary abnormality [14]. Renal alterations in magnesium and calcium reabsorption could result from a functional or structural defect in the DCT caused by the loss of NCC activity rather than as a consequence of hypokalemia or alkalosis. In turn, magnesium depletion promotes urinary potassium excretion, favoring hypokalemia [15]. Conversely, in BS, in which hypomagnesemia is mild or absent, the upregulation of the TRPM6 expression is a compensatory mechanism in order to restore the reduced reabsorption of Mg^2+^ in the thick ascending limb (TAL) [16]. The exact target for plasma magnesium in GS and BS patients is uncertain. A level >0.6 mmol/L is considered to be reasonable [4].

Correction of hypomagnesemia should be done by diet and, above all, by oral supplements [4].

Dietary magnesium is present in vegetal and animal foods. Legumes, nuts, green leafy vegetables, seeds, bananas, whole grains, dark chocolate, and fish and are good sources of this mineral (Table 2). Food processing, such as refining grains, reduces the magnesium content [17] and therefore is preferable whole grains.

Administration of oral supplements is recommended as a primary treatment not only to correct hypomagnesemia but also to prevent chondrocalcinosis [2,18,19] and favor potassium absorption [20].

Magnesium supplementation is often poorly tolerated due to abdominal pain and diarrhea induced by high doses of magnesium salts [21]. Few data are available on the bioavailability of different magnesium supplementation salts [22]. Animal studies showed that organic salts (e.g., aspartate, citrate, fumarate, gluconate, and lactate) are more bioavailable than inorganic forms (e.g., oxide, hydroxide, and chloride) and, among organic salts, magnesium gluconate exhibits the highest bioavailability [23]. 

It is recommended to start magnesium supplementation with a dose of 300 mg/day (12.24 mmol) of elemental magnesium (5 mg/kg in children, i.e., 0.2 mmol/kg) using, if possible, slow-release tablets [2]. Magnesium supplements should be taken in small frequent doses (3–4 times/day) and with meals in order to reduce/avoid the chance of diarrhea that may worsen hypokalemia and volume depletion [24].

To increase magnesium bioavailability, liposomes have been employed both for drug and nutrients delivery [25]. Liposomes, vesicles consisting of one or more phospholipid bilayers surrounding an aqueous solution core, are able to cross the cell membrane, carrying their contents inside the cells [26]. In the field of nutrient supplementation, liposomes are employed for vitamin and mineral delivery [25]. Recently, a new delivery system consisting of phospholipids covered by a sucrester matrix has been proposed both for iron and magnesium supplementation. Sucrester is a surfactant derived from the esterification of fatty acids with sucrose (sucrose esters) that has been shown to act as an absorption enhancer due to its ability to reduce intestinal barrier resistance [27]. 

Sucrosomial magnesium has demonstrated a higher bioavailability both in vitro and in human studies compared to others magnesium forms. A commercial preparation of sucrosomial magnesium was compared to other forms of magnesium dietary supplements [28]. Ex vivo evaluation showed that sucrosomial magnesium was absorbed faster and in larger amounts in respect to magnesium oxide, while in healthy subjects, surosomial preparation induced higher magnesium concentrations than citrate, oxide, and bisglycinate forms in plasma, urine, and red blood cells. 

Sucrosomial magnesium is absorbed in the intestine via a passive paracellular mechanism, allowing to bypass the concentration gradient that would limit its absorption (Figure 2).

Abbreviations: ATP, adenosine triphosphate and TRP M6/7, transient receptor potential melastatin type 6 and 7 magnesium channels.

In the sucrosomial form, magnesium is protected from the interaction with other substances that may limit its adsorption. Due to this mechanism, the amount of magnesium not absorbed in the intestine is lower and the absorbed magnesium protected by sucrosome exerts a low osmotic effect, reducing the risk of diarrhea [28]. However, until now, clinical studies concerning the effectiveness of sucrosomial magnesium in restoring its plasma levels are lacking. Preliminary data from our laboratory shows in a cohort of hypomagnesemic renal transplanted patients, a higher effectiveness and tolerability of sucrosomial magnesium (ULTRAMAG^®^ PharmaNutra S.p.A, Pisa, Italy) compared to magnesium pidolate (Innico et al. 2021, abstract accepted at the Italian Society of Nephrology-SIN-annual meeting, Rimini, Italy, October 2021).

As mentioned above, a similar technology has been used for iron supplementation, evidencing that sucrosomial iron was more tolerated and more effective compared to other forms of magnesium [29]. For example, high doses of oral sucrosomial iron were as effective as intravenous iron in terms of increasing blood hemoglobin levels with the advantage of a lower cost [30,31].

Proton pump inhibitors (PPI) impair the intestinal magnesium absorption, acting on magnesium transporters in particular in genetically predisposed individuals [32]; thus, patients with hypomagnesemia should discontinue PPI and switch to alternative medical treatments such as ranitidine or famotidine and dietary treatments [33].

When oral supplementations are not able to improve hypomagnesemia or in the case of severe muscle cramps, weekly intravenous magnesium sulfate may be used [2]. 

In GS patients with hypomagnesemia, vitamin D levels should also be evaluated. Magnesium, in fact, plays a central role in several steps of the vitamin D metabolism [34] and in particular in the enzymatic conversion of 25(OH)D3 to the active form 1,25(OH)D3, which is a magnesium-dependent process [35].

### 2.4. Food Causing Potassium and Magnesium Loss

GS patients should consider that some foods may cause a loss of potassium and/or magnesium. Licorice root contains glycyrrhizinic acid, an inhibitor of the enzyme 11β-hydroxysteroid dehydrogenase type 2 that converts cortisol into cortisone [36]. Cortisol, unlike cortisone, exerts mineralocorticoid activity, inducing sodium retention and increasing potassium excretion. A prolonged use of products containing glycyrrhizinic acid (above 50 mg daily), including herbal preparations, dietary supplements, candies, and liqueurs, may cause hypokaliemia, sodium and water retention (with or without hypertension), and rhabdomyolysis [37]. Usually, rhabdomyolysis can occur when potassium levels fall below 2.0 mEq/L [38]. Cases of rhabdomyolysis induced by licorice in patients with GS has been reported [39].

GS patients should avoid excessive use of alcoholic beverages. Indeed, alcohol abuse may result in electrolyte disorders including hypomagnesemia and hypokalemia [40]. These abnormalities are mainly due to tubular dysfunctions [41]. Electrolyte alterations may occur both in the chronic abuse and binging of alcohol consumption [42], and persistent hypokalemia with episodes of muscle weakness have been reported following alcohol abuse in GS patients [43].

Some beverages, such as fruit juice, citrate, and bicarbonate-rich beverages, as well as almond-based beverages, favor metabolic alkalosis and induce a further fall in plasma potassium levels. Fruit juices may cause alkalosis due to the presence of organic acids, mainly citrate and malate. These organic acids are an intermediate of the Krebs cycle and once adsorbed are mostly oxidized [44]. Moreover, citrate is converted to bicarbonate predominantly in the liver [45], worsening metabolic alkalosis [46].

Bicarbonate and bicarbonate-rich beverages, such as sparkling waters, may have a strong alkalinizing effect [47]. Almond-based beverages can induce alkalosis and hypokalemia in children when they totally or partially replaced cow’s milk or infant formula probably because of the low chloride content [48].

GS and BS patients should limit fruit juice intake considering both their organic acids and sugar content. As an alternative, a moderate amount (<500 g/d) of fresh fruit which has a higher content of potassium and less carbohydrates is preferable (see Table 2).

## 3. Conclusions

The dietary approach to BS and GS can be done through sodium, potassium, and magnesium-rich foods and/or supplements containing these minerals. In the case of poor tolerance to magnesium supplements, more tolerable liposomal (sucrosomial) formulation can be used.

Some foods and beverages can contribute to reducing plasma potassium and magnesium levels and therefore these patients or their parents, in the case of children with these syndromes, should know them and moderate the use of these beverages and foods. However, for all the reasons reported and considered in this review, in addition to the regular nephrologic follow-up visits and since the very beginning of the evaluation of the treatment options, a nutritional consultation is certainly useful for an appropriate nutritional approach for the treatment of GS and BS patients. It should be also made available to the parents of GS and BS, these latter mostly children, and a nutritionist should regularly join the nephrologist in the follow-up care of GS and BS patients.

## Figures and Tables

**Figure 1 nutrients-13-02960-f001:**
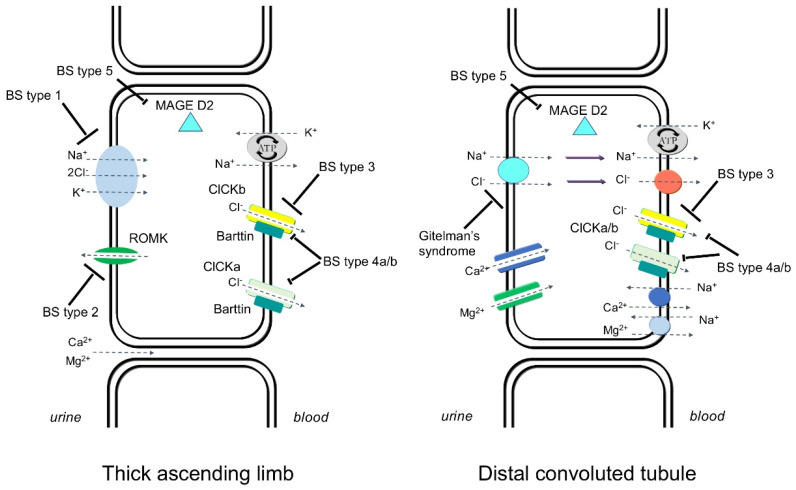
Transport pathways in the thick ascending limb of Henle’s loop and in the distal convoluted tubule depicting the abnormalities of the five types of Bartter’s syndrome and Gitelman’s syndromes.

**Figure 2 nutrients-13-02960-f002:**
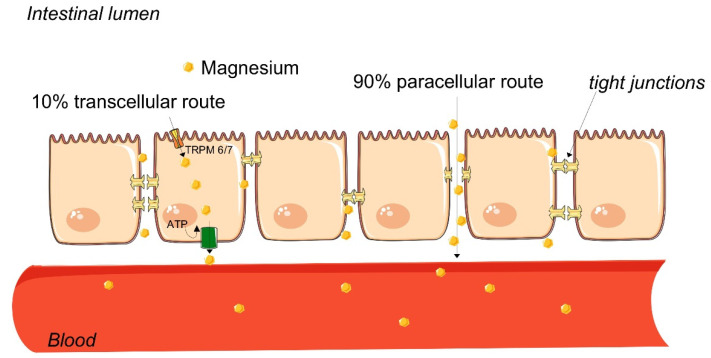
Sucrosomial magnesium intestinal absorption. Magnesium ions encapsulated within a sucrosomial membrane pass through the intestine without direct interaction with the mucosa and lumen content, and then are absorbed via passive paracellular mechanisms that account for about 90% of the whole intestinal uptake. Only a small part of magnesium (about 10%) is absorbed through membrane transporters (TRP M6/7).

**Table 1 nutrients-13-02960-t001:** Foods rich in potassium (content for 100 g of an edible portion).

Food	Potassium (mg)	Carbohydrates (g)	Energy (Kilocalories)
*Vegetables*			
Broccoli, raw	303	6.3 (sugars 1.4)	39
Kale, raw	348	4.4 (sugars 0.8)	43
Lettuce, raw	253	3 (sugars 1.2)	20
Mushrooms (white), raw	318	3.3 (sugars 2.0)	22
Spinach, raw	460	3.6 (sugars 0.4)	23
Tomatoes, canned	198	7.3 (sugars 4.4)	32
Potatoes, boiled	372	20.1 (sugars 0.9)	87
Swiss chard, raw	379	3.7 (sugars 1.1)	19
*Legumes*			
Beans (white), dry	1540	60.3 (sugars 2.1)	333
Green peas, raw	244	14.4 (sugars 5.7)	81
Soy milk	158	14.4 (sugars 5.7)	38
*Fruits*			
Apricot, raw	259	11.1 (sugars 9.2)	48
Apricot, dried	1162	62.6 (sugars 53.4)	241
Avocados, raw	485	8.6 (sugars 0.3)	167
Banana, raw	326	8.6 (sugars 0.3)	89
Kiwifruit, green, raw	312	14.7 (sugars 9.0)	61
Orange, raw	166	11.5 (sugars 9.1)	46
Watermelon	173	7.5 (sugars 6.2)	30
Grapefruit juice	141	7.7 (sugars 7.7)	37
Fruit juice blend, 100% fruit	101	11.2 (sugars 10.4)	46
*Nuts*			
Almonds	733	21.6 (sugars 4.3)	579
Hazelnuts	680	17.7 (sugars 4.3)	628
Walnuts	441	13.7 (sugars 2.61)	654
*Meat and fish*			
Halibut (Atlantic and Pacific), raw	435	0	91
Salmon (Atlantic, farmed), raw	363	0	208
Beef (short loin), raw	266	0	145
Pork (shoulder, fresh), raw	302	0	236
*Other*			
Chocolate, dark 60–69%	567	52.4 (sugars 36.7)	579
Cocoa powder	1524	57.8 (sugars 1.75)	228

Data obtained from the site of the U.S. Department of Agriculture: https://fdc.nal.usda.gov// accessed on 17 August 2021.

**Table 2 nutrients-13-02960-t002:** Foods rich in magnesium (content for 100 g of an edible portion).

Food	Magnesium (mg)	Carbohydrates (g)	Energy (Kilocalories)
*Nuts*			
Almonds	270	21.6 (sugars 4.3)	579
Hazelnuts	163	17.7 (sugars 4.3)	628
Walnuts	158	13.7 (sugars 2.6)	654
*Legumes*			
Beans (black), dry	180	40.8 (sugars 2.4)	270
Chickpeas, dry	45	25.5 (sugars 4.5)	210
Green peas, raw	33	14.4 (sugars 5.7)	81
Lentils, raw	33	63.4 (sugars 2.0)	352
*Cereals and starchy foods*			
Bran flakes	229	80.5 (sugars 18.6)	328
Quinoa, uncooked	197	64.2 (sugars 2.7)	368
Quinoa, cooked	64	21.3 (sugars 0.9)	120
Wheat germ, crude	239	51.8 (sugars 16.8)	360
Whole bread	69	55.9 (sugars 2.9)	262
*Vegetables*			
Spinach, raw	79	3.6 (sugars 0.4))	23
Swiss card, raw	81	3.7 (sugars 1.1)	19
*Others*			
Avocado	29	8.6 (sugars 0.3)	167
Chocolate, dark 60–69%	176	52.4 (sugars 36.7)	579
Chocolate, dark 70–85%	228	45.9 (sugars 0)	598
Mackerel, raw	60	0	205
Tofu	30	2.35 (sugars 24)	71
Tofu, yogurt	40	16 (sugars 1.2)	94

Data obtained from the site of the U.S.Department of Agriculture: https://fdc.nal.usda.gov/. accessed on 17 August 2021.

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
