# Peer review of "The Dietary Approach to the Treatment of the Rare Genetic Tubulopathies Gitelman’s and Bartter’s Syndromes"

_nutrients, 2021, doi:10.3390/nu13092960_

Round 1
Reviewer 1 Report
Gitelman and Bartter syndromes are two settings characterized by a renal losses of sodium, potassium and magnesium. In this review, the authors aimed to focus on the dietary strategies to increase input of these ions by either foods or supplements.
General comment:
The utility to separate Gitelman and Bartter syndromes is unclear since, as the authors notice, sentences 249-251, BS presents clinical characteristics virtually indistinguishable from GS and it dietary treatment largely shares the same intervention. Perhaps would it be better to discuss successively sodium, potassium and magnesium intakes and in each paragraph, the way to increase intakes according to the needs. Indeed, the distinction between these syndromes is that in antenatal and néonatal BS but not in Bartter 3 or GS, the needs in sodium and potassium are quite elevated and cannot be provided by food.
Introduction
1/ The description of Gitelman and Bartter syndromes, including genetic aspects, is not mandatory to discuss the way to increase input of sodium, potassium and magnesium. It can be shortly presented, before discussing their treatment and appropriate dietary needs (sentences 87-102).
2/Potassium content of the food : the authors mention the” differences between meat and vegetable as source of potassium/or magnesium” in terms of calories and glucids content . However these contents are not provided in tables 2 and 3.
3/Oral supplements of magnesium: this is an important point: The bioavaibility of salts should be indicated. Biovalabily of magnesium oxide is reported as low as 2% while that of Mg pidolate at about 50%
4/The comparators of sucrosomial magnesium should be discussed (reference 22). Is this preparation available for clinical practice? Lines 204 : how many hypomagnesemic transplanted patients were studied? What was the design (crossover?)
Minor comments :
1/Table, 2, potassium content of cacao beverage is fivefold higher than pure cacao that is quite surprising…
2/Table 2 and 3: the sort order of the foods is unclear : not alphabetical nor by K or Mg content….
3/Perhaps give the equivalence to mmol of KCl as mg, because oral supplements
4Line 151-153 Mechanism of hypomagnesemia in GS was studied By Nijenhuis et al (JClin Invest 2005). They showed during chronic HCTZ administration and in NCC-knockout mice, an animal model of Gitelman syndrome, a downregulation of the epithelial Mg2+ channel transient receptor potential channel subfamily. This may provide on explanation to GS-related hypomagnesemia. Hypomagnesemia is mild in BS. It has be experimentally demonstrated Increased expression of renal TRPM6 compensates for Mg2+ wasting during furosemide treatment (Angel et Al-CKJ 2012)
Author Response
We thank the Reviewer for his/her appropriate comments that helped us to improve our paper.
As for specific comments:
Comments and Suggestions for Authors
Gitelman and Bartter syndromes are two settings characterized by a renal losses of sodium, potassium and magnesium. In this review, the authors aimed to focus on the dietary strategies to increase input of these ions by either foods or supplements.
General comment:
The utility to separate Gitelman and Bartter syndromes is unclear since, as the authors notice, sentences 249-251, BS presents clinical characteristics virtually indistinguishable from GS and it dietary treatment largely shares the same intervention. Perhaps would it be better to discuss successively sodium, potassium and magnesium intakes and in each paragraph, the way to increase intakes according to the needs. Indeed, the distinction between these syndromes is that in antenatal and néonatal BS but not in Bartter 3 or GS, the needs in sodium and potassium are quite elevated and cannot be provided by food.
Introduction
1/ The description of Gitelman and Bartter syndromes, including genetic aspects, is not mandatory to discuss the way to increase input of sodium, potassium and magnesium. It can be shortly presented, before discussing their treatment and appropriate dietary needs (sentences 87-102).
The most relevant clinical aspects of the syndromes have been gathered in the same paragraph as suggested and removed the table that summarizes the genetic defects of the syndromes.
2/Potassium content of the food : the authors mention the” differences between meat and vegetable as source of potassium/or magnesium” in terms of calories and glucids content. However these contents are not provided in tables 2 and 3.
Energy and carbohydrates content have been added in the related tables, tables 1 and 2 of the revised version of the manuscript.
3/Oral supplements of magnesium: this is an important point: The bioavaibility of salts should be indicated. Biovalabily of magnesium oxide is reported as low as 2% while that of Mg pidolate at about 50%
Data about the bioavailability of magnesium forms vary depending on the experimental models and the study design. Despite this, organic forms (i.e., citrate, pidolate) result always more bioavailable, as we have reported in the manuscript.
4/The comparators of sucrosomial magnesium should be discussed (reference 22). Is this preparation available for clinical practice?
We have briefly discussed the data of reference 22 that now is listed as [28].
Lines 204 : how many hypomagnesemic transplanted patients were studied? What was the design (crossover?)
The study we have performed has considered 19 patients with sucrosomial magnesium (ULTRAMAG® PharmaNutra S.p.A, Pisa, Italy) and 13 patients with magnesium pidolate. The design was a pilot parallel study performed in a single center.
Minor comments :
1.Table, 2, potassium content of cacao beverage is fivefold higher than pure cacao that is quite surprising…
We have replaced cocoa beverage and pure cocoa with dark chocolate and cocoa powder.
- Table 2 and 3: the sort order of the foods is unclear: not alphabetical nor by K or Mg content….
Both tables were sorted by kind of food and by alphabets.
- Perhaps give the equivalence to mmol of KCl as mg, because oral supplements
Equivalence mmol/mg for potassium and magnesium was added in table 4.
- Line 151-153 Mechanism of hypomagnesemia in GS was studied By Nijenhuis et al (JClin Invest 2005). They showed during chronic HCTZ administration and in NCC-knockout mice, an animal model of Gitelman syndrome, a downregulation of the epithelial Mg2+ channel transient receptor potential channel subfamily. This may provide on explanation to GS-related hypomagnesemia. Hypomagnesemia is mild in BS. It has be experimentally demonstrated Increased expression of renal TRPM6 compensates for Mg2+ wasting during furosemide treatment (Angel et Al-CKJ 2012)
We have added these reports in the text.

Reviewer 2 Report
- For Table 2 and Table 3, suggest to be more organized by either ranking by the amount of potassium and magnesium from highest to lowest; or by alphabets.
- Any intersections of food that have both high in potassium and high in magnesium that patients can choose;
- Any special meal plates that patients can choose that have high in potassium and magnesium
- This sentence "Supplements based on the liposomal and in particular sucrosomial technology, while having at least the same efficacy on magnesium level, could reduce the adverse effects, as previously reported" need reference and also need to be more specific. What is generic name or trade name of this formula? Is it available in U.S. or just in Europe? Over the counter?
- "Some foods and beverages can contribute to reduce plasma potassium and magnesium levels" What type of food? Can you please be more specific?
- Some beverages favor metabolic alkalosis and induce a further fall in plasma potassium level. Is this real? Can you please give examples? and references?
- Fruit juices may cause alkalosis? Which Fruit juices? and how much is too much? Given these fruits and vegetables are also recommended given they have high potassium and magnesium contents.
Reviewer 3 Report
Francini et al conducted a detailed review on the dietary approach to the treatment of the rare genetic tubulopathies Gitelman’s and Bartter’s syndromes. The manuscript is well written and very informative.
Comments:
1.I suggest to add a table resuming the recommended starting doses of Na, K and Magnesium supplements for each disease as well as a table resuming the dietary approaches that should be avoided in these diseases.
2.Moreover, a comment on the need for tube feeding (or by gastrostomy) in some young patients with need for high doses of sodium chloride and potassium supplements in order to maximize caloric intake and assure optimal growth needs to be added.
Round 2
Reviewer 1 Report
The revised version adequately answer to my initial review. The manuscript is clear, well documented.
Reviewer 3 Report
I have no further comments